# Impacts of cellulase and xylanase addition on antibiotic resistance and microbial community during dairy manure composting

**Ping Gong, Yuan Zhou, Daoyu Gao, Pingmin Wan, Zhiyong Shao, Erguang Jin** *

Institute of Animal Husbandry and Veterinary, Wuhan Academy of Agricultural Science, Wuhan, Hubei, P.R.China

* jeg0617@126.com

## Abstract

### Background

Composting is a transformation and biodegradation process that converts organic biomass into valuable products while also removing antimicrobial resistance genes (ARGs). Promoting lignocellulose biodegradation is essential for enhancing composting efficiency and improving the quality of compost derived from agricultural organic waste. This study aims to explore the effects of cellulase and xylanase on the composting process of cow manure, with a focus on their impact on key physicochemical properties, microbial communities, and antibiotic resistance genes (ARGs).

### Methods

Dairy manure compost was carried out for 30 days with cellulase and xylanase treatment. The physicochemical characteristics (pH, organic matter (OM), total nitrogen (TN), available nitrogen (AN), germination index (GI), humic acid (HA), and fulvic acid (FA)) of the compost samples were measured, along with enzymatic activities, including cellulase activity (CA), urease activity (UA), alkaline phosphatase (ALP), and dehydrogenase activity (DHA). Furthermore, bacterial communities and ARGs were analyzed using 16S rRNA gene sequencing and high-throughput quantitative PCR. Additionally, network properties, redundancy analysis, and variation partitioning analysis were conducted.

### Results

Enzyme additions significantly enhanced composting efficiency, which improved temperature regulation and increased nitrogen content. Cellulase notably accelerated the degradation of organic matter, enhanced microbial diversity, and reduced ARG abundance, while xylanase played a crucial role in stabilizing pH and temperature during the later stages, facilitating nitrogen retention and compost maturity. Additionally,

**Data availability statement:** The high-throughput sequence data have been deposited in the National Center for Biotechnology Information (NCBI) BioProject database with project number PRJNA1246353 (https://www.ncbi.nlm.nih.gov/bioproject/?term=PRJNA1246353). All other relevant data are within the paper and its Supporting Information files.

**Funding:** This work was supported by the Tibet Autonomous Region Key R&D Project (XZ201801NB34) and the Wuhan Academy of Agricultural Sciences Scientific and Technological Innovation Project (XTCX2021002). There was no additional external funding received for this study. The funders had no role in study design, data collection and analysis, decision to publish, or preparation of the manuscript.

**Competing interests:** The authors have declared that no competing interests exist.

microbial community dynamics were closely linked to ARG patterns, indicating that enzymatic treatments can optimize composting processes while mitigating the spread of resistance genes.

## Conclusion

The findings highlight the complementary roles of these enzymes in improving composting outcomes and suggest strategies for sustainable waste management. These findings provide valuable insights for improving the composting efficiency and quality of compost derived from agricultural organic waste.

## Introduction

Antibiotic resistance genes (ARGs) have been recognized as a world widely concerned environmental pollutant which posed increasingly severe risks to human health, controlling and eliminating ARGs in various environments has become one of the emerging frontiers [1]. The widespread use of antibiotics in livestock industry has led to the accumulation and spread of diverse ARGs and mobile genetic elements (MGEs) in animal manure. [2–4]. With the rapid development of the dairy industry, large-scale intensive livestock farming generates significant amounts of manure, which has become one of the major sources of agricultural pollution. [5]. The use of manure as fertilizer and/or improper treatment methods can spread abundant ARGs in soil, which may then enter the food chain through plant endophytic bacteria, posing a threat to human health [6]. Therefore, it is crucial to develop effective ways in reducing risks of ARG dissemination in livestock manure, particularly for high-risk ARGs with clinical significance as identified by the World Health Organization. [7].

At present, composting is currently a promising harmless treatment technology for livestock manure, capable to remove the antibiotics and pathogenic microorganisms in feces [8]. As a widely used technique in livestock manure management, composting offers advantages such as environmental friendliness, low energy consumption, and high cost-effectiveness [9]. Pathogenic bacteria and drug-resistant microorganisms in the manure will be effectively killed due to high temperatures during composting, which will also accelerate the degrading of most antibiotics present in animal waste [10,11]. Additionally, composting can effectively mitigate the complex contamination of antibiotics and drug-resistant genes in manure by improving microbial composition and structure, which in turn reshape the diversity and abundance of ARGs [12,13]. Enhanced correlations between horizontal gene transfer and bacterial communities could promote the persistence of ARGs at high antibiotic levels during composting [14]. Therefore, changes and relations in bacterial community and ARG abundance during composting are critical factors in the effective and harmless treatment of manure.

In addition to killing pathogenic microorganisms and removing ARGs, composting is a transformation and biodegradation process which converts organic biomass into value added products. The microbiome, as the intrinsic drivers of biotransformation

during composting, plays a critical role in driving the process either independently or synergistically [15]. Lignocellulose is the most abundant but stable organic compound in composts, and the recalcitrant nature of lignocellulose hinders the utilization of cellulose and hemicellulose, decreasing the bioconversion efficiency of lignocellulose [16]. Therefore, promoting the biodegradation of lignocellulose is of great significance to improving composting process [17]. Microbes secreting significant quantities of lignocellulase competent to hydrolyze lignocellulose during composting, but the diversity and quantity were enough in composting samples [16,17]. Cellulase and xylanase were effect on degrading cellulose and hemicellulose [16], while the changes in microbial community and ARG during dairy manure composting have not been studied.

In this study, cellulase and xylanase were individually added to laboratory-scale dairy manure composting. The antibiotic resistome, microbial community, and several physicochemical properties of the compost samples were analyzed. Objectives of this study were to evaluate the effects of cellulase and xylanase on the physicochemical properties, microbial community, and antibiotic resistome.

## Materials and methods

Fresh dairy cattle manure and fermentation bed were collected from a cattle farm in Wuhan, China. The fresh dairy cattle manure was stool and urine mixture with about 80% water content and fermentation bed was consist of rice husk and sawdust. Basic properties of raw materials for fermentation are shown in Table 1. The compound microbial agent was mixed of *Bacillus subtilis*, *Enterococcus faecalis*, *Lactobacillus* and *Saccharomyces*, and total bacterial content ≥$1 \times 10^{10}$ CFU/g. Compound microbial agent, xylanase ($2 \times 10^5$ U/g) and cellulase ($8 \times 10^3$ U/g), provided by Wuhan Xurun Environmental Protection Technology Co., LTD.

The composting trial was conducted using lidless plastic containers placed under a rain shelter at the dairy farm. Each plastic box had a total volume of 0.7 m³ (height: 765 mm, upper diameter: 1060 mm, lower diameter: 950 mm). Twelve small holes (1 cm in diameter) were made at the bottom, and nine holes of the same size were distributed along the sides. The total volume of the bedding material was 0.5 m³, composed of a mixture of rice husks and sawdust in a 3:2 volume ratio. To each mixture, 100 g of a compound microbial agent, 120 kg of fresh manure, and 1.3 kg of urea were added. Additionally, for the treatment groups, 50 g of enzyme preparation was incorporated into each batch The moisture content was adjusted to 60%, and the carbon-to-nitrogen (C/N) ratio to 30:1. After thoroughly mixing, the prepared material was placed in the plastic containers for composting. The experiment was divided into three groups, namely compound microbial agent (Group A), compound microbial agent and cellulase (Group B), compound microbial agent and xylanase (Group C). The experimental groups are detailed in Table 2. A real-time digital thermometer was inserted vertically into the center of the compost at a depth of 20–30 cm to monitor the temperature. The temperature of the digital thermometer and the air

**Table 1. Basic properties of rice husk, sawdust and cow manure.**

| Ingredient | Carbon (g/kg) | Nitrogen (g/kg) | C/N (g/kg) | Water content (%) |
|---|---|---|---|---|
| rice husk | 360 | 4.8 | 75 | 10 |
| sawdust | 593.2 | 2.6 | 288.2 | 20 |
| cow manure | 318 | 13.3 | 23.91 | 80 |

**Table 2. Group setting of the experiment.**

| Groups | Treat |
|---|---|
| A | 100 g compound microbial agent |
| B | 100 g compound microbial agent + 50 g cellulase |
| C | 100 g compound microbial agent + 50 g xylanase |

temperature and humidity measured by the air temperature-humidity meter were recorded daily at 9:00 a.m. The composting process lasted for 30 days, with turning operations performed on days 0, 3, 10, and 20.

Samples were collected on days 0, 3, 10, 20, and 30 using the five-point sampling method from the top, middle, and bottom layers (approximately 200 mm, 400 mm, and 600 mm from the surface). The collected samples were further processed using the quartering method. All samples were labeled with group information and collection dates, and stored in a −20°C freezer for future use.

## Analysis of physicochemical parameters

Fresh samples collected from composting pad were used for pH, organic matter (OM), total nitrogen (TN) and available nitrogen (AN) using previously described methods [18] based on Chinese National Agricultural Organic Fertilizer Standard (NY525–2012) [19]. The germination index (GI) was determined according to the previous study [20]. Values of humic acid (HA) and fulvic acid (FA) were determined based on previous research [21].

## Analysis of enzyme activity

Cellulase activity (CA), Urease activity (UA), alkaline phosphatase (ALP), dehydrogenase activities (DHA) were determined by Cellulase Activity Assay Kit (BC0155, Solarbio, Beijing, China), Urease Activity Assay Kit (BC0125), Alkaline Phosphatase Activity Assay Kit (BC0280) and Dehydrogenase Activity Assay Kit (BC0395), respectively. All measurements were performed according to the manufacturer's instructions.

## DNA extraction and high-throughput qPCR

Total DNA was extracted from 0.25 g of compost sample by Power Soil DNA Isolation Kit (MoBio Laborataries Inc. CA, USA). The concentration and purity of DNA samples were measured using a micro-volume spectrophotometer (NanoDrop ND-1000, Nano Drop Technologies Inc., DE, USA). The extracted DNA samples showed A260/A280 ratios between 1.8 and 2.0. The DNA samples were uniformly diluted to 20 ng/µL using sterile water and stored at −20°C for future testing.

High-throughput quantitative PCR (HT-qPCR) platform (SmartChip Real-time PCR system, Wafergen Inc., USA) was employed to quantitate the abundance of antibiotic resistance genes (ARGs) as described in previous study with slightly modified [11,15]. A total of 296 pairs of primers (Supporting information S1) selected for this study were effectively validated in previous studies, including 285 pairs for detecting antibiotic resistance genes, 10 pairs for detecting mobile genetic elements (MGEs), and 1 pair for detecting 16S rDNA. After the reaction, qPCR data were exported based on the pre-set screening criteria of the Cycler (amplification efficiency between 1.8 and 2.2 with a single peak in the melting curve). The detection limit of the instrument was established at a CT value of 31 according to the performance specifications of the SmartChip Real-Time PCR Systems. Each sample was subjected to three technical replicates, and amplification was considered positive if all three replicates yielded amplification.

## Microbial analysis

Bacterial 16S rRNA gene was amplified using primers 341F (5'-ACT CCT ACG GGA GGC AGC AG-3') and 806R (5'-GGA CTACHV GGG TWT CTA AT-3'). The PCR conditions were pre-denaturation at 98°C for 2 min, 25 cycles of denaturation at 98°C for 15 s, annealing at 55°C for 30 s, elongation at 72°C for 30 s, and a final post-elongation cycle at 72°C for 5 min. The PCR products were purified with AMPure XP beads (AXYGEN). After purification, the PCR products were used for the construction of libraries and then paired-end sequenced on Illumina Miseq (Illumina, CA, USA) at the Bioyigene, Wuhan, China. Microbiome bioinformatics were performed with QIIME 2 2019.4 [22] with slight modification according to the official tutorials (https://docs.qiime2.org/2019.4/tutorials/).

## Network analysis

Microbiota sequencing data from composting samples was used to perform network analysis based on previous work [18] with slight modification. Briefly, absolute abundance data were used to analyze correlation at genus level within group. Relative abundances of the 200 most abundant species were selected. SparCC correlation coefficients > 0.85 and false discovery rate-corrected *P*-values < 0.01 were defined as robust correlations. Topology parameters was calculated by Network Analyzer in Cytoscape 3.10.1 [23].

## Statistical analysis

Statistical assessment of physiochemical properties, enzymes activity and alpha diversity was performed by two-way ANOVA followed by post hoc test using Tukey's multiple comparisons test. GI was assessed by ordinary one-way ANOVA followed by post hoc test using Tukey's multiple comparisons test. Topological properties were assessed by Kruskal-Waillis test with Dunn's multiple comparison test. In the figures, $P < 0.05$ indicates statistical significance (*$P < 0.05$, **$P < 0.01$, ***$P < 0.001$). Analyses were performed using R (R Core Team, Vienna, Austria) and Graphpad Prism (version 9.3.0, Graphpad Software Inc, La Jolla, California, USA).

## Results

According to the temperature changes during the composting process, the composting process can be divided into two stages, namely the heating stage (day 1–7) and the cooling stage (day 8–30) (Fig 1A). In the thermophilic phase, the maximum temperature of group A remained above 50°C for about 3 days, and group B reached around 50 °C on the second day and remained for about 7 days, although there is a small fluctuation (Fig 1A). Besides, the maximum temperature

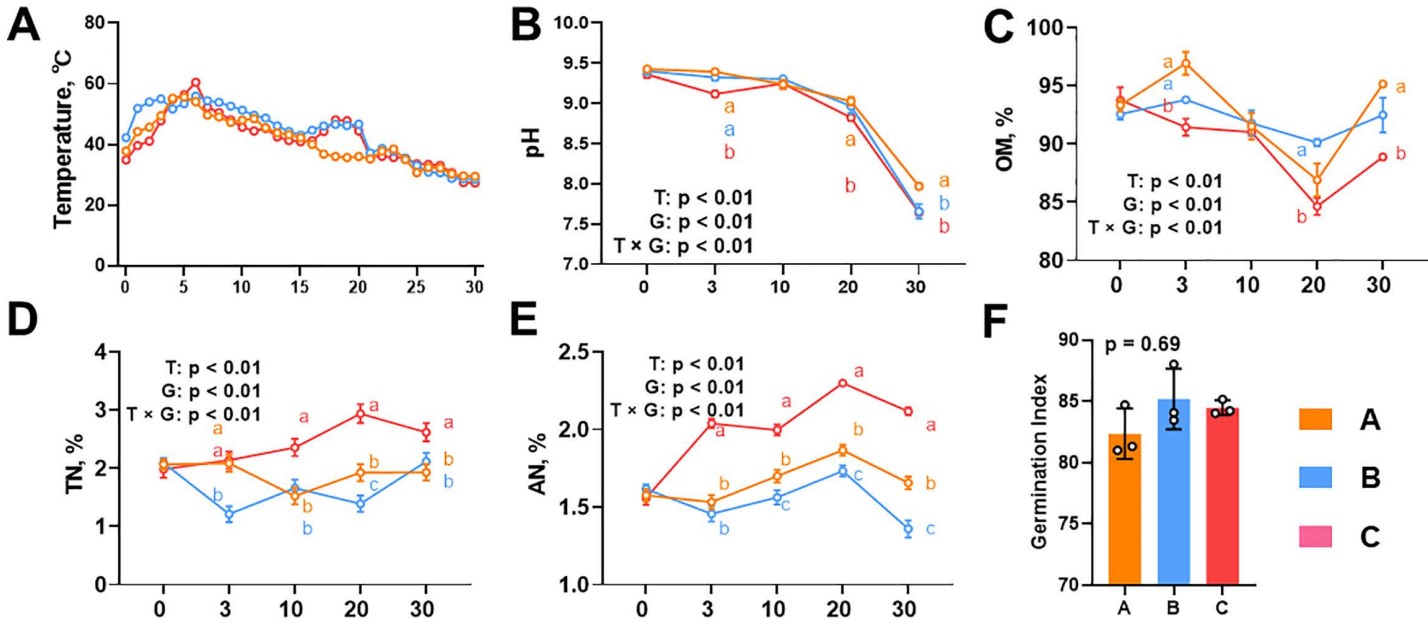

**Fig 1. Physio-chemical properties changes during dairy manure composting.** Temperature (A), pH (B), organic matters (C), total nitrogen (D), available nitrogen (E) and germination index (F) were assayed during the dairy manure composting process. Ordinary two-way ANOVA with Tukey's multiple comparisons test was used for statistical assessment of Temperature, pH, organic matters, total nitrogen and available nitrogen. Ordinary one-way ANOVA followed by post hoc test using Tukey's multiple comparisons test was used for statistical assessment of germination index. Values with different letters are significantly different at *p* < 0.05. OM, organic matters; TN, total nitrogen; AN, available nitrogen.

of group C reached the maximum temperature about 60 °C at day 7 (Fig 1A). The pH of three treatments showed similar variation tendencies, which maintained between 9.0–9.5 in 20 days and sharply decreased from 9.0 to about 7.5–8.0 in later 10 days, while group B and C showed lower pH compared to group A at day 30 (Fig 1B). OM increased at day 3 for group A and B, followed a decrease to day 20, then increased (Fig 1C). Meanwhile, OM of group C at day 3 and 30 were significantly lower than group A (Fig 1C). As to TN and available N, group C showed highest values during composting process (Fig 1D and 1E). On the other side, TN and available N gradually increased during composting process while other two group showed some fluctuates (Fig 1D and 1E). The germination index showed on difference of the three treatments (Fig 1F). These data indicated addition of cellulase and xylanase enhanced composting efficiency.

The values of FA during the composting process showed increasing, while group B showed highest value and group C showed lowest (Fig 2A). During composting, the content of HA increased in the whole time, but the three different treatments showed different changing pattens (Fig 2B). HA content of group A rose continuously at early followed by a decrease at day 20, and increased again. HA content of group B experienced a sharp decline in 0–3 days and rapid increase in the following week, while remained almost unchanged after 10 days. HA content of group C showed a fluctuating increase (Fig 2B). Change of CA content in three groups varied during the whole process (Fig 2C). Content of CA in group A and C increased significantly in the early three days and decreased in the following week. After, content of CA in group A then continued to rise, while it in group C continued to rise followed a decline (Fig 2C). The CA content of group B remained stable for the first three days followed a sharply increased, which then experienced decreasing and increasing (Fig 2C). UA content in the three groups showed same trend during the composting process, namely increasing on 0–3 day, decreasing on 3–10 day followed an increase on day 10–30 (Fig 2D). The ALP of group A and B basically unchanged in the early three day followed an increase and then decrease, while ALP of group A reached peak value at day10 and that

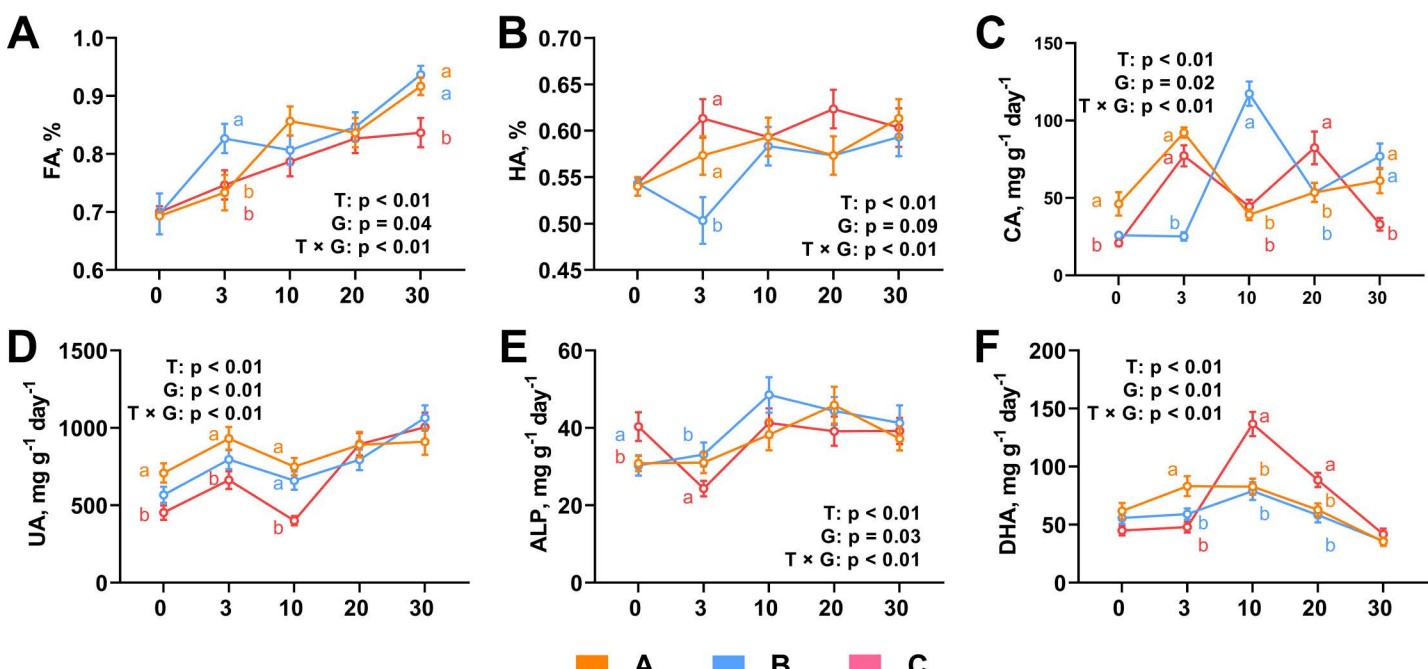

**Fig 2. Enzymes activity changes during dairy manure composting.** Fulvic acid (A), humic acid (B), cellulase activity (C), urease activity (D), alkaline phosphatase (E) and dehydrogenase activities (F) were assayed during the dairy manure composting process. Ordinary two-way ANOVA with Tukey's multiple comparisons test was used for statistical assessment. Values with different letters are significantly different at $p < 0.05$. FA, fulvic acid; HA, humic acid, CA, cellulase activity, UA, urease activity, ALP, alkaline phosphatase; DHA, dehydrogenase activities.

of group B reached at day 20 (Fig 2E). ALP of group C fluctuated greater than group A and B during composting (Fig 2E). As for DHA, there was little change in group A and C in the early 10 days, followed a significant decrease in 10–30 day (Fig 2F). The DHA of group C rose abruptly to peak value from day 3 to day 10, and then gradually decreased (Fig 2F). There was no difference in the final content of UA, ALP and DHA at the end of composting among three groups (Fig 2D–2F). The addition of cellulase and xylanase enhanced the decomposition of organic matter and nutrient conversion during the composting process.

Bacterial alpha diversity was decreased gradually (Fig 3A–3D) (T: $p < 0.01$). On the other side, addition of cellulase and xylanase significantly increased the Pielou_e and Shannon index (Fig 3C and 3D, G: $p < 0.01$). Specifically, addition of cellulase increased Chao1 index and Faith_pd index at day 3 (Fig 3A and 3B), while xylanase enhanced Pielou_e index and Shannon index after composting 3 days (Fig 3C and 3D). At the end of experimental time (day 30), cellulase and xylanase increased the Pielou_e index and Shannon index (Fig 3C and 3D). Based on principal co-ordinates analysis (PCoA) showed in Fig 3E, cellulase and xylanase changed the beta diversity as compost continue. Samples located together at day 0 and gradually showed separation as compost proceeding. Microbial diversity varied mainly at day 3 for adding cellulase, while that seem happened day 10 for xylanase. After day 10, microbial construction variation seems to be stable. Cellulase and xylanase significantly enhanced microbial community diversity, which is crucial for optimizing composting processes. Their effects were observed at early (day 3) and later (day 10) stages, respectively, promoting both the richness and evenness of the microbial community.

During composting, obvious temporal changes of microbial compositions were observed at the phylum level (Fig 4B). *Firmicutes*, *Proteobacteria*, *Bacteroidetes* and *Actinobacteria* were the top four dominant phyla, which accounting for 84.31% − 98.86%. *Firmicutes* increased lightly in group A and C followed a rapidly decline, and that displayed drastic fluctuation in group B (Fig 4A). On the other side, *Proterbacteria* decreased in early stage and increased later in group A and C, while it increased dramatically at day 3 in group B. *Bacteroidetes* boosted after 20- or 30-day composting in three

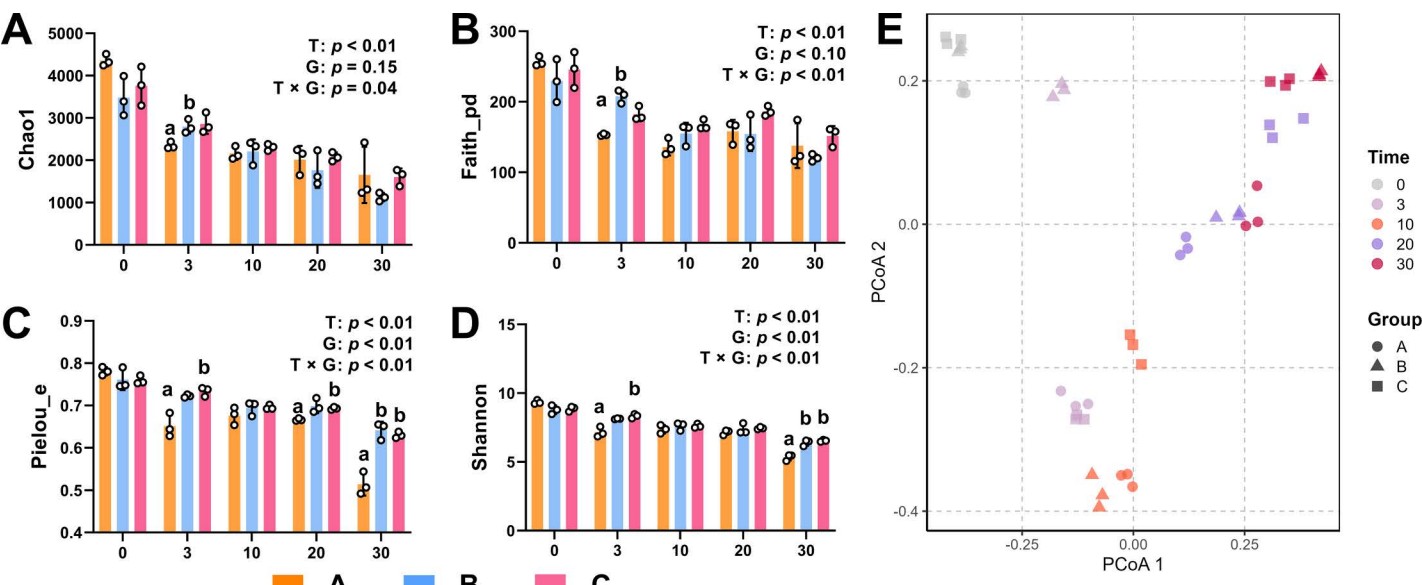

**Fig 3. Microbial diversity changes during dairy manure composting.** Chao1 (A), Faith_pd (B), Pielou_e (C), shannon (D) and Principal coordinate analysis (E) were assayed during the dairy manure composting process. Ordinary two-way ANOVA with Tukey's multiple comparisons test was used for statistical assessment. Values with different letters are significantly different at $p < 0.05$.

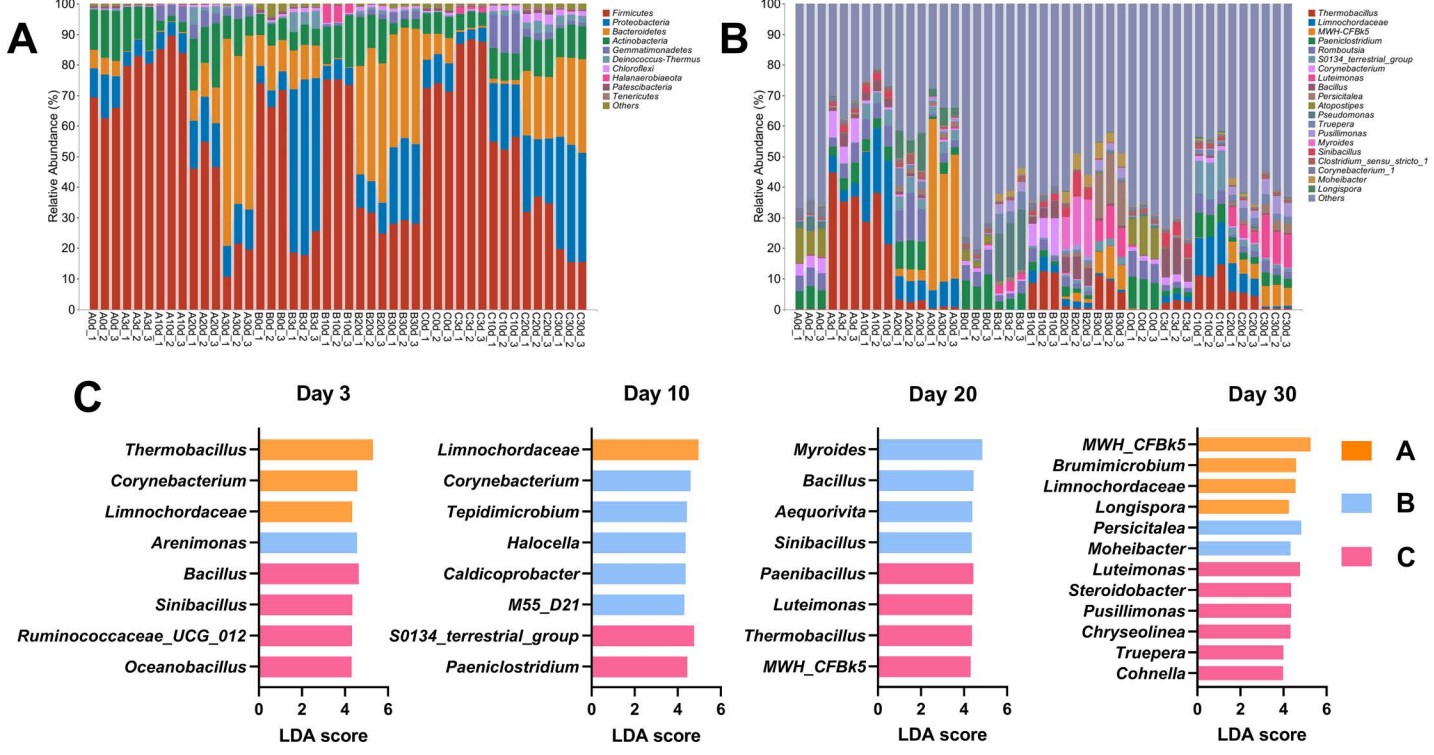

**Fig 4. Changes of microbial composition dairy manure composting.** Relative abundance of the top 10 phyla (A) and top 20 genera (B). LEfSe analysis showed biomarkers in different phases (C). Biomarkers were showed with LDA score＞4.3.

groups. As for *Actinobacteria*, group A displayed decreased followed by increased while that in group B and C showed reversed tendency.

At genus level, *Paeniclostridium*, *Romboutsia* and *Atopostipes* were dominant at day 0, and the relative abundance were 17.91%, 21.61% and 26.84%, respectively. With composting, the relative abundance of *Thermobacillus* showed a brief increase and subsequent decrease. The relative abundance of *Thermobacillus* in group A samples increased to 38.90% after 3-day composting with a slight decrease in the following week, while dropped to less than 1% in the later composting period (Fig 4B). Besides, the relative abundance of *Thermobacillus* in the group B samples increased to 10.99% at day10 followed a decrease, and then increased to 8.65% after day30 (Fig 4B). After a brief increase in the relative abundance of *Thermobacillus* in group C, it gradually decreased to the state before composting (Fig 4B).

Based on LEfSe results (Fig 4C), after 3-day composted, cellulase added group (group B) enriched *Arenimonas*, and xylanase added group (group C) enriched *Bacillus*, *Sinibacillus*, *Ruminococcaceae_UCG_012* and *Oceanobacillus*. At day 10, cellulase added group (group B) enriched *Corynebacterium*, *Tepidimicrobium*, *Halocella*, *Caldicoprobacter* and *M55_D21*, while *S0134_terrestrial_group* and *Paeniclostridium* were enriched in xylanase added group (group C). At 20 days, *Myroides*, *Bacillus*, *Aequorivita* and *Sinibacillus* were promoted in group B samples. while *Paenibacillus*, *Luteimonas*, *Thermobacillus* and *MWH_CFBk5* were enriched in xylanase added group (group C). At day 30, *Persicitalea* and *Moheibacter* were enriched in cellulase added group (group B) and *Luteimonas*, *Steroidobacter*, *Pusillimonas*, *Chryseolinea*, *Truepera* and *Cohnella* were biomarkers for xylanase added group (group C).

A total of 282 antibiotic resistance genes (ARGs) were identified across the collected samples (Fig 5A and 5B). The detected ARGs conferred resistance to a wide range of antibiotics, with the most prevalent being multidrug resistance (19.08%), beta-lactamase (16.60%), macrolide-lincosamide-streptogramin B (MLSB) (16.14%), aminoglycoside (15.41%),

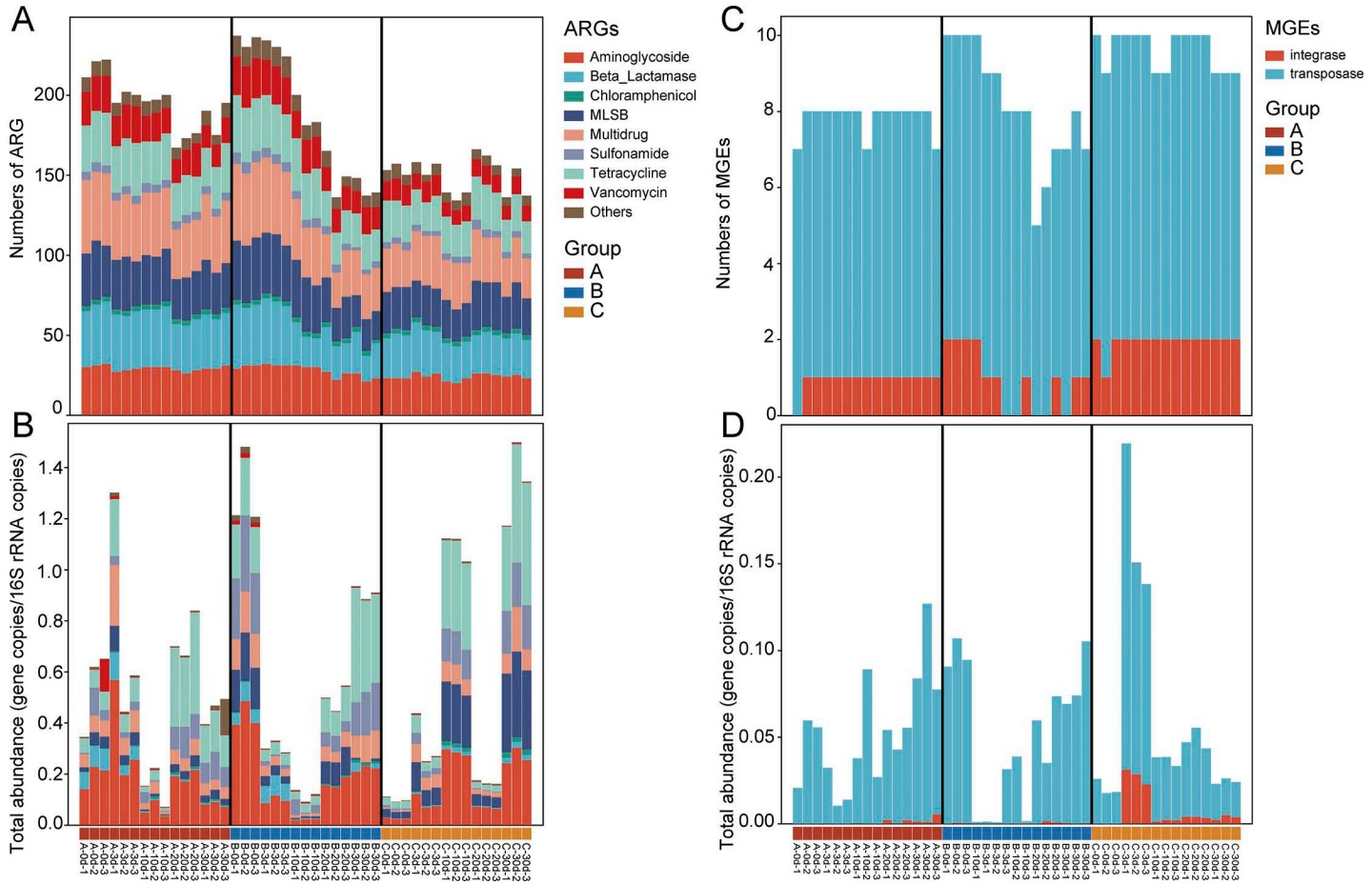

**Fig 5. Changes of ARGs and MGEs during dairy manure composting.** (A) The average number and (B) the total abundances of ARGs. (C) The average number and (D) the total abundances of MGEs.

and tetracycline (14.76%). The predominant mechanism contributing to antibiotic resistance was antibiotic deactivation (41.67%), followed by the action of efflux pumps (30.51%) and cellular protection (24.72%). The composting process significantly reduced ARG diversity across all three groups, with reductions ranging from 7.17% to 39.69%. Notably, the most pronounced decrease was observed in Group B samples after 20 days of composting (Fig 5A). Furthermore, the total abundance of ARGs varied among the three groups throughout the composting period. In Group A, the total abundance of ARGs initially declined but showed recovery after 10 days, followed by a slight decrease from day 20 to day 30 (Fig 5B). Group B exhibited a decrease in ARG abundance during the first 10 days, but subsequently increased during the maturation stage (Fig 5B). In Group C, the total abundance of ARGs dramatically decreased after an initial rise at day 10, followed by a significant increase from day 20 to day 30 (Fig 5B). As to MGEs, total numbers and integrase showed slightly decrease in group B (Fig 5C). Total abundance of MGEs in group A and B both reduced at day 3 followed an increased while the variation tendency of group C opposite to group A and B (Fig 5D). These data underscore cellulase and xylanase improved efficacy of composting in reducing ARG diversity.

The bacterial co-occurrence network indicated the effect of enzymes addition on microbial community during composting. As seen in Fig 6 and Table 3, the bacterial community changed after composting with cellulase (Group B) and xylanase (Group C) addition compared to control (Fig 6A–6C). Cellulase (Group B) and xylanase (Group C) treatment increased

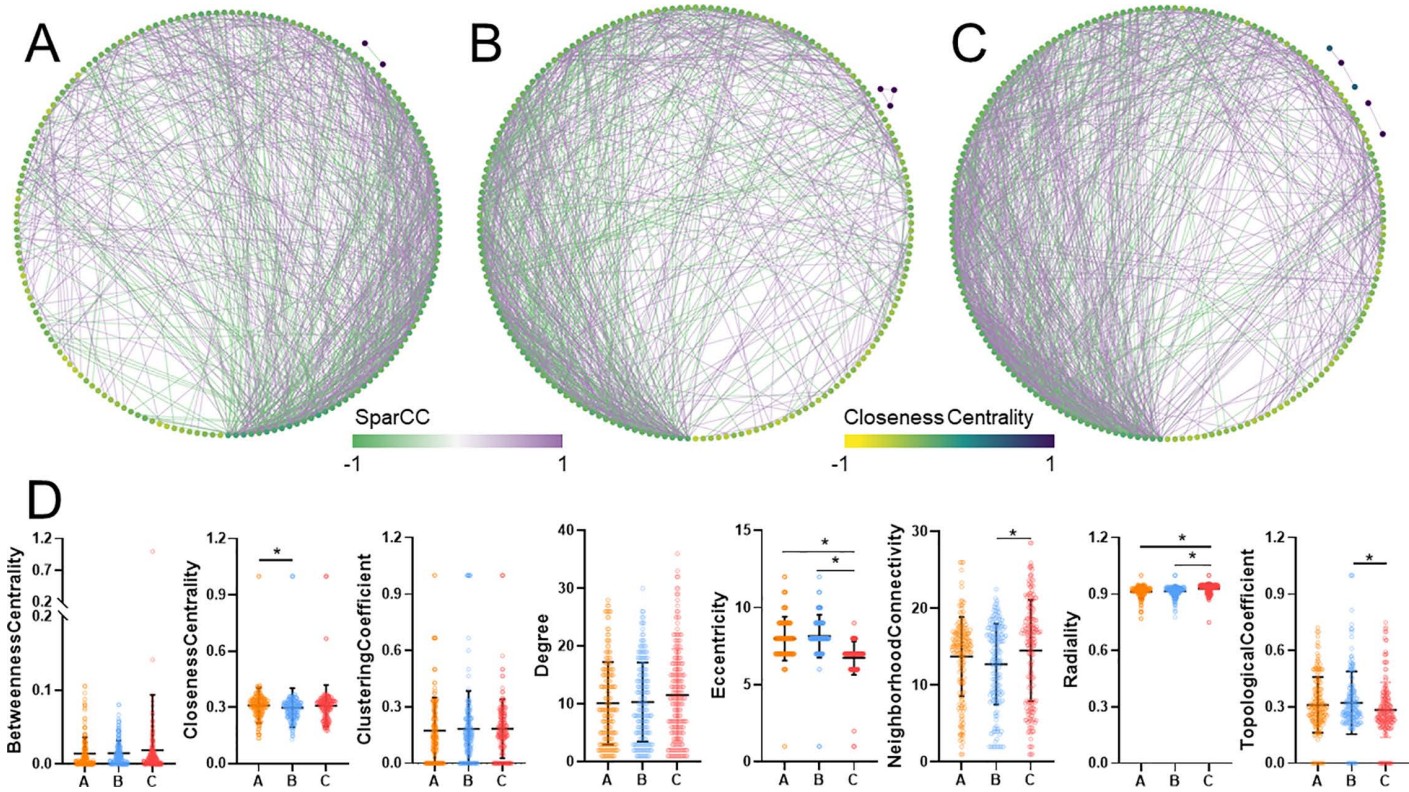

**Fig 6. Network analysis and topological properties of bacterial community during dairy manure composting.** In bacterial network of control (A), cellulase (B) and xylanase (C), nodes in the network represent taxon (species level), and node color is proportional to closeness centrality; green edge indicates negative correlation and purple edge indicates positive correlation. Topological properties of nodes in bacterial networks (D). Kruskal-Waillis test with Dunn's multiple comparison test was used for statistical assessment of topological properties. Values with asterisk are significantly different at $p < 0.05$.

**Table 3. Network topological properties of bacterial community during dairy manure ectopic fermentation[1].**

|  | A | B | C |
|---|---|---|---|
| Numbers of nodes | 173 | 179 | 187 |
| Number of edges | 875 | 927 | 1077 |
| Ave number of neighbors | 10.222 | 10.500 | 11.802 |
| Network diameter | 12 | 12 | 9 |
| Network radius | 6 | 6 | 6 |
| Characteristic path length | 3.459 | 3.624 | 3.533 |
| Clustering coefficient | 0.175 | 0.170 | 0.189 |
| Network density | 0.060 | 0.060 | 0.065 |
| Network heterogeneity | 0.693 | 0.646 | 0.670 |
| Network centralization | 0.106 | 0.113 | 0.135 |
| Connected components | 2 | 2 | 3 |

[1]Network node and edge data was imported to cytoscape and microbial network properties was analyzed by network analyzer module.

Number of edges, Ave number of neighbors and Network centralization (Table 3). On the other hand, Group C showed lower Closeness Centrality, Eccntricity and Topological Coefficient, while had higher Neighborhood Connectivity and Radiality based on node properties (Fig 6D). These data indicated that addition of cellulase (Group B) and xylanase (Group C) resulted in elevated microbial community interaction and enhanced microbial community complexity during composting.

Redundancy analysis (RDA) was used to analyze the links among physio-chemical properties, bacterial community and ARG profiles in compost (Fig 7A and 7B). The contributors of physio-chemical properties to the microbial community varied in different phases of composting. *Firmicutes*, which were dominant in the early stage of composting, mainly contributed to the CA, OM and pH (Fig 7A). *Bacteroidetes*, which were dominant in the later stage, mainly contributed to the HA, UA and FA (Fig 7A). On other hand, *Bacteroidetes* responded to cellulase addition while *Proteobacteria* and | *Deinococcus-Thermus* contributed to xylanase addition (Fig 7A). Contributors of microbial community to the ARG patterns also varied in different phases of composting. *Firmicutes* mainly influenced day 0 to day 10 and contributed to Vancomycin (Fig 7B). xylanase and *Proteobacteria* was most correlated with MLSB, Aminoglycoside, Multidrug and Transposase, while cellulase addition and *Bacteroidetes* was correlated with Sulfonamide, Chloramphenicol and Tetracycline (Fig 7B).

VPA analysis was conducted to evaluate further the influences of bacteria community and environmental factors on the ARGs and MGEs profiles. Results indicated that the factors explained 80.32% of the total variation in the ARGs and MGEs profiles (Fig 8). Among them, the joint effect showed the greatest contribution (40.24%) to the ARG profiles, followed by bacterial community (28.21%) and physio-chemical properties (11.86%).

## Discussion

This study examines the impact of cellulase and xylanase on the composting process, highlighting temperature changes, pH levels, nitrogen content, and microbial dynamics. The composting phases included a heating phase (days 1–7) and a

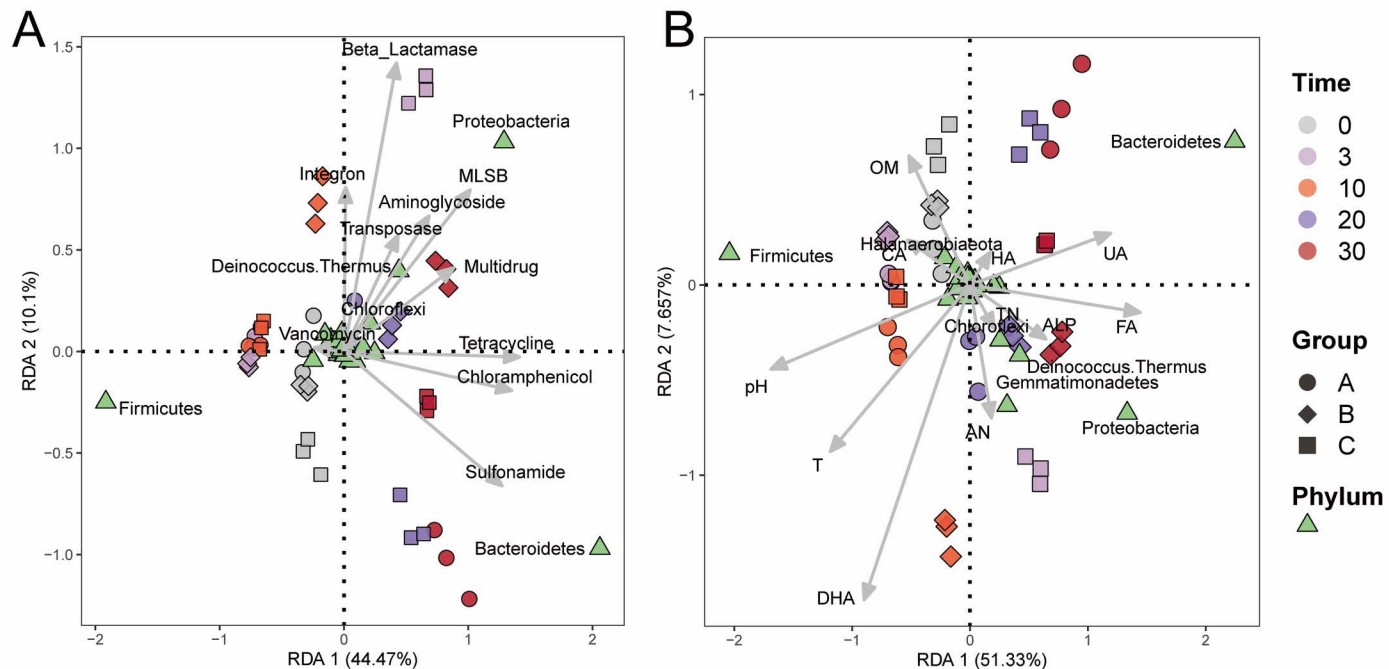

**Fig 7. Redundancy analysis (RDA) among bacteria, physiochemical properties and ARG and MGE.** RDA was used to assess the relations of bacterial communities with ARG and MGE(A), as well as with environmental factors (B) during the dairy manure composting. The values of axes are the percentages explained by the corresponding axis.

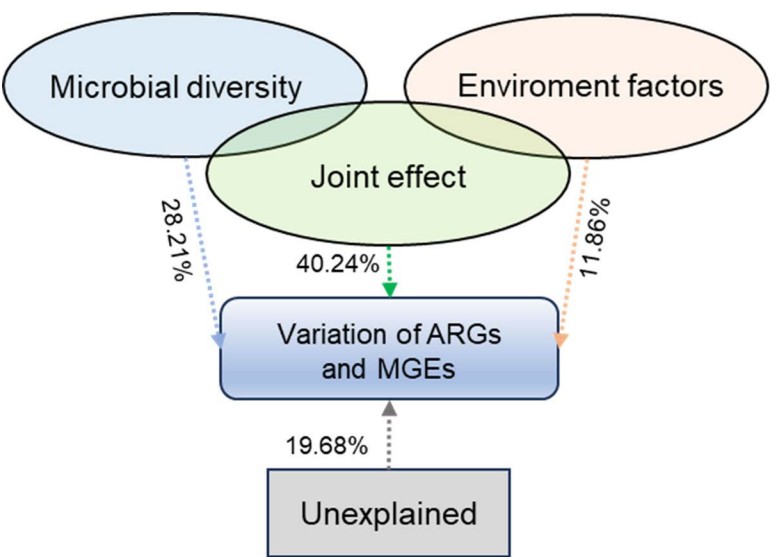

**Fig 8. Variation partitioning analysis (VPA) revealing the contributions of environmental factors and microbial community to the changes of ARGs and MGEs.**

cooling phase (days 8–30), with notable variations in microbial diversity and activity due to enzyme addition. Furthermore, the composting process effectively reduced ARG diversity. Redundancy analysis and co-occurrence network analysis provided insights into the relationships between environmental factors and microbial communities, contributing to optimized composting practices.

The addition of cellulase and xylanase significantly enhanced the composting process through distinct mechanisms. Evidenced by improved temperature maintenance, pH variation, and nitrogen content, cellulase exhibited superior thermophilic activity and acidification characteristics, while xylanase showed greater potential in temperature regulation and nitrogen accumulation. Cellulase accelerated the decomposition of organic matter by promoting a rapid temperature rise and prolonging the thermophilic phase, leading to early acidification and increased nitrogen release. This rapid degradation of organic matter shortened the composting cycle and improved nitrogen availability [24,25]. On the other hand, xylanase played a critical role in stabilizing temperature and pH levels during the later stages of composting, contributing to sustained nitrogen retention and greater compost maturity. This stabilization enhances the long-term quality of the compost [26]. In addition, both enzymes influenced the formation of humic substances differently. Cellulase induced early humic acid accumulation, while xylanase promoted more fluctuating changes in humic substances, ensuring comprehensive organic matter transformation across the composting stages [27]. This is in line with this study that xylanase treatment resulted in greater fluctuations in the decomposition of organic matter and nutrient conversion within the composting system. Furthermore, microbial diversity was enhanced by both cellulase and xylanase, with cellulase promoting early-stage microbial activity and xylanase supporting network stability and diversity in the later stages [28]. Together, cellulase and xylanase complement each other by optimizing microbial interactions, nutrient cycling, and system stability, making them valuable tools for enhancing the efficiency and quality of composting processes [29–31].

Microbial diversity decreased during compost process and addition of the two enzymes boosted diversity of microbial community. Cellulase and xylanase play critical roles in modulating microbial communities during composting by accelerating the decomposition of cellulose, hemicellulose, and xylan [32]. These polysaccharides represent key components of plant-derived organic matter, and their enzymatic breakdown releases oligosaccharides and other intermediates that support microbial metabolism [33]. Studies show that cellulase promotes thermophilic microbial

activity, encouraging taxa specialized in the degradation of complex carbohydrates, particularly *Firmicutes* and thermophilic genera like *Thermobacillus* [34,35]. This can enhance microbial diversity at high temperatures, facilitating organic matter turnover and nitrogen cycling in early composting stages [34,35]. Xylanase influences microbial communities differently, as its breakdown products (e.g., xylose) preferentially stimulate bacterial taxa associated with nitrogen fixation and secondary metabolism, such as *Bacillus* and *Paenibacillus* [35,36]. These shifts can enhance nutrient retention, especially nitrogen, improving compost quality. Enzymatic treatments also foster greater microbial interaction and stability, reflected in increased microbial network complexity, which is crucial for maintaining ecosystem functionality throughout the composting process [37,38]. The differential effects of cellulase and xylanase indicate their complementary roles: cellulase facilitates rapid organic matter turnover, while xylanase promotes more stable community dynamics and nutrient conservation. These findings align with the broader literature emphasizing that targeted enzyme additions can improve composting efficiency by accelerating decomposition and modulating microbial networks [38–41]. The strategic use of these enzymes in composting systems offers potential for optimizing microbial community composition, accelerating maturation, and enhancing compost quality.

The addition of cellulase and xylanase during composting markedly influences the dynamics of antibiotic resistance genes (ARGs) and microbial community structures. Cellulase enhances the degradation of complex organic matter, promoting microbial diversity and facilitating ARG reduction. Conversely, the current study revealed that xylanase results in significant fluctuations in microbial communities and metabolic products, with a notable increase in ARG abundance during the later composting stages. This indicated that xylanase generates readily assimilable sugars, stimulating specific microbial groups that may harbor ARGs [42]. These fluctuations highlight the dual nature of enzyme applications; while enhancing composting efficiency, they may inadvertently promote the survival of resistant microorganisms [16,34]. Therefore, managing enzyme types and levels is critical for balancing composting efficiency and ARG mitigation. Further research is needed to understand the intricate relationships between enzymatic activity, microbial metabolism, and resistance gene dynamics, which are essential for developing sustainable composting strategies [4,10].

This study highlights the significant roles of cellulase and xylanase in optimizing the composting process. Looking ahead, identifying key microorganisms that respond positively to enzymatic treatments is crucial. Combining targeted enzyme formulations with selected microbial consortia can further enhance composting efficiency and sustainability. This synergistic approach aligns with the concept of "One Health," promoting not only improved waste management but also environmental and public health. Continued research in this area is essential for developing effective, sustainable composting strategies that optimize efficiency while minimizing the risks associated with antibiotic resistance.

## Conclusion

Addition of cellulase and xylanase during composting both enhanced composting efficiency according to nutrition turnover, microbial community change and microbial interactions, although their differential effects are notable. Cellulase accelerates organic matter decomposition, enhancing temperature rise and nitrogen release, thus promoting microbial diversity and reducing antibiotic resistance genes (ARGs). On the other side, xylanase stabilizes pH and temperature in the later stages, contributing to nitrogen retention and compost maturity.

## Supporting information

**S1 File. Primer list used in high-throughput quantitative PCR (HT-qPCR).**
(DOCX)

**S2 File. Source data for figures.** This file contains the raw numerical data used to generate all figures presented in the main manuscript.
(XLSX)

## Acknowledgments

We thank Institute of Animal Husbandry and Veterinary, Wuhan Academy of Agricultural Science and Bioyigene (China) for their assistance.

## Author contributions

**Conceptualization:** Ping Gong, Erguang Jin.

**Data curation:** Yuan Zhou.

**Formal analysis:** Ping Gong.

**Funding acquisition:** Erguang Jin.

**Investigation:** Daoyu Gao, Pingmin Wan.

**Methodology:** Yuan Zhou, Zhiyong Shao.

**Project administration:** Erguang Jin.

**Resources:** Pingmin Wan.

**Supervision:** Erguang Jin.

**Validation:** Yuan Zhou.

**Visualization:** Daoyu Gao.

**Writing – original draft:** Ping Gong.

**Writing – review & editing:** Ping Gong, Yuan Zhou, Daoyu Gao, Pingmin Wan, Zhiyong Shao, Erguang Jin.

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
