## [Decision Letter · Decision Letter 0]

10 Mar 2025

Dear Dr. Jin,

Thank you for submitting your manuscript to PLOS ONE. After careful consideration, we feel that it has merit but does not fully meet PLOS ONE’s publication criteria as it currently stands. Therefore, we invite you to submit a revised version of the manuscript that addresses the points raised during the review process.

**Dear Authors,**

**Your manuscript, "Impacts of Cellulase and Xylanase Addition on Antibiotic Resistance and Microbial Community During Dairy Manure Composting," has been reviewed by two reviewers. One reviewer has suggested major revisions, while the other has recommended minor revisions. Please address the comments accordingly to improve the quality of your manuscript.**

**Best regards,**

We look forward to receiving your revised manuscript.

Kind regards,

Shweta Sharma, PhD

Academic Editor

PLOS ONE

**Journal Requirements:**

Please ensure that your manuscript meets PLOS ONE's style requirements, including those for file naming. The PLOS ONE style templates can be found at https://journals.plos.org/plosone/s/file?id=wjVg/PLOSOne_formatting_sample_main_body.pdf and https://journals.plos.org/plosone/s/file?id=ba62/PLOSOne_formatting_sample_title_authors_affiliations.pdf 2. Thank you for stating in your Funding Statement: This work was supported by the Tibet Autonomous Region Key R&D Project (XZ201801NB34) and the Wuhan Academy of Agricultural Sciences Scientific and Technological Innovation Project (XTCX2021002). Please provide an amended statement that declares *all* the funding or sources of support (whether external or internal to your organization) received during this study, as detailed online in our guide for authors at http://journals.plos.org/plosone/s/submit-now.  Please also include the statement “There was no additional external funding received for this study.” in your updated Funding Statement. Please include your amended Funding Statement within your cover letter. We will change the online submission form on your behalf. 3. In the online submission form, you indicated that The original contributions presented in the study can be contacted to the corresponding author.All PLOS journals now require all data underlying the findings described in their manuscript to be freely available to other researchers, either a. In a public repository, b. Within the manuscript itself, or c. Uploaded as supplementary information.This policy applies to all data except where public deposition would breach compliance with the protocol approved by your research ethics board. If your data cannot be made publicly available for ethical or legal reasons (e.g., public availability would compromise patient privacy), please explain your reasons on resubmission and your exemption request will be escalated for approval. 4. When completing the data availability statement of the submission form, you indicated that you will make your data available on acceptance. We strongly recommend all authors decide on a data sharing plan before acceptance, as the process can be lengthy and hold up publication timelines. Please note that, though access restrictions are acceptable now, your entire data will need to be made freely accessible if your manuscript is accepted for publication. This policy applies to all data except where public deposition would breach compliance with the protocol approved by your research ethics board. If you are unable to adhere to our open data policy, please kindly revise your statement to explain your reasoning and we will seek the editor's input on an exemption. Please be assured that, once you have provided your new statement, the assessment of your exemption will not hold up the peer review process. 5. Please amend either the abstract on the online submission form (via Edit Submission) or the abstract in the manuscript so that they are identical.

**Additional Editor Comments:**

Dear Erguang Jin,

Your submission Impacts of cellulase and xylanase addition on antibiotic resistance and microbial community during dairy manure composting has been reviewed and we would like to encourage you to submit revisions that address the reviewers' comments. An editor will review these revisions and if they address the concerns adequately, your submission may be accepted for publication.

The reviewers' comments are included at the bottom of this email. Please respond to each point in the reviewers' comments and identify what changes you have made. If you find any of the reviewer's comments to be unjustified or inappropriate, please explain your perspective.

We look forward to receiving your revised submission.

Kind regards,

Editor, PLOS One

Reviewers' comments:

Reviewer's Responses to Questions

**Comments to the Author**

1. Is the manuscript technically sound, and do the data support the conclusions?

Reviewer #1: Yes

Reviewer #2: Yes

2. Has the statistical analysis been performed appropriately and rigorously?

Reviewer #1: Yes

Reviewer #2: Yes

3. Have the authors made all data underlying the findings in their manuscript fully available?

Reviewer #1: Yes

Reviewer #2: Yes

4. Is the manuscript presented in an intelligible fashion and written in standard English?

Reviewer #1: Yes

Reviewer #2: Yes

**Reviewer #1: ** Comments to Authors:

Major points

1- In results: you wrote many sentences as discussion!!! The rule of

manuscripts’ writing is to avoid using any discussing points in the results

chapter, so you should transfer those sentences to discussion part.

2- You should add conclusion part at the end of discussion chapter.

Minor points

1- In this manuscript; you wrote (5 times Our)!!! The rule of manuscript

writing is to avoid using (Our). So you should delete (Our) and use academic

scientific words such as (This study or The current study or The present

study).

Kind regards

**Reviewer #2: ** Thanks for all Researchers for this research psper. The paper is well written and acceptable for publication after addressing some comments

More details needed on Mann whiteny test used to assess differece of what

tukey used in anova need to be added in the Figure legend

**Do you want your identity to be public for this peer review?** For information about this choice, including consent withdrawal, please see our Privacy Policy

Reviewer #1: No

Reviewer #2: No

---

## [Author Response · Author response to Decision Letter 1]

9 Apr 2025

Response to reviewers, PONE-D- 24-53884

Dear Editor,

We are grateful for the opportunity to revise our manuscript entitled “Impacts of cellulase and xylanase addition on antibiotic resistance and microbial community during dairy manure composting” in light of the editorial and reviewer comments. Below, we have provided a point-by-point response.

Academic editor's comments

1.When submitting your revision, we need you to address these additional requirements. Please ensure that your manuscript meets PLOS ONE's style requirements, including those for file naming. The PLOS ONE style templates can be found at https://journals.plos.org/plosone/s/file?id=wjVg/PLOSOne_formatting_sample_main_body.pdf and https://journals.plos.org/plosone/s/file?id=ba62/PLOSOne_formatting_sample_title_authors_affiliations.pdf

Response:

We have ensured that our manuscript meets PLOS ONE's style requirements.

2.Thank you for stating in your Funding Statement:

This work was supported by the Tibet Autonomous Region Key R&D Project (XZ201801NB34) and the Wuhan Academy of Agricultural Sciences Scientific and Technological Innovation Project (XTCX2021002).

Response:

We have provided the following amended Funding Statement:

“This work was supported by the Tibet Autonomous Region Key R&D Project (XZ201801NB34) and the Wuhan Academy of Agricultural Sciences Scientific and Technological Innovation Project (XTCX2021002). There was no additional external funding received for this study. The funders had no role in study design, data collection and analysis, decision to publish, or preparation of the manuscript.”

Please see line 451-456 in manuscript.

3.In the online submission form, you indicated that The original contributions presented in the study can be contacted to the corresponding author.

Response:

We have provided the data in an Excel uploaded as “Supporting Information files.xlsx”, all relevant data except for sequence data are in the file.

4.When completing the data availability statement of the submission form, you indicated that you will make your data available on acceptance. We strongly recommend all authors decide on a data sharing plan before acceptance, as the process can be lengthy and hold up publication timelines. Please note that, though access restrictions are acceptable now, your entire data will need to be made freely accessible if your manuscript is accepted for publication. This policy applies to all data except where public deposition would breach compliance with the protocol approved by your research ethics board. If you are unable to adhere to our open data policy, please kindly revise your statement to explain your reasoning and we will seek the editor's input on an exemption. Please be assured that, once you have provided your new statement, the assessment of your exemption will not hold up the peer review process.

Response:

We have uploaded our 16S-DNA sequence data to NCBI and revised the “Data Availability Statement” section. Please see Line 435-439.

5.Please amend either the abstract on the online submission form (via Edit Submission) or the abstract in the manuscript so that they are identical.

Response:

We have carefully re-read and revised our manuscript, and we didn’t revise the abstract of manuscript.

Response to Reviewer #1

We sincerely appreciate your time and effort in reviewing our manuscript. Your insightful comments and constructive suggestions have been invaluable in improving the quality and clarity of our work. We are grateful for your thoughtful feedback and have carefully addressed each point in our revised manuscript.

Major points

1- In results: you wrote many sentences as discussion!!! The rule of manuscripts’ writing is to avoid using any discussing points in the results chapter, so you should transfer those sentences to discussion part.

Response: Thank you for your insightful comment. We appreciate your guidance on the proper structure of the manuscript. Upon reviewing the Results section, we identified and relocated sentences that contained interpretative or discussion-like content to the Discussion section. The Results section now strictly presents the findings without interpretation. We have revised the manuscript accordingly, and the changes can be found at line 242, line 314-316, line 318-320, line 357-360, line 371-373, line 379-380.

2- You should add conclusion part at the end of discussion chapter.

Response: Thank you for your valuable suggestion. We agree that a conclusion section would enhance the clarity and impact of our manuscript. Accordingly, we have added a conclusion at the end of the Discussion section, summarizing the key findings and their implications. The revised manuscript now includes this section at line 427-434.

Minor points

1- In this manuscript; you wrote (5 times Our)!!! The rule of manuscript writing is to avoid using (Our). So you should delete (Our) and use academic scientific words such as (This study or The current study or The present study).

Response: Thank you for your careful review and valuable suggestion. We acknowledge that using more formal academic expressions enhances the clarity and professionalism of the manuscript. Accordingly, we have replaced instances of "Our" with more appropriate terms such as "This study," "The current study," or "The present study" throughout the manuscript. These revisions can be found at line 91-92, line 318, line 371 and line 407.

We appreciate your insightful feedback, which has helped us improve the quality of our manuscript.

Response to Reviewer #2

We sincerely appreciate your thorough review of our manuscript. Your thoughtful feedback and detailed suggestions have greatly contributed to improving the clarity and rigor of our work. We highly value your time and effort in evaluating our study, and we have carefully considered and addressed all your comments in our revision.

Comment: More details needed on Mann whiteny test used to assess differece of whattukey used in anova need to be added in the Figure legend

Response: Thank you for your positive feedback and for recognizing the quality of our manuscript. We appreciate your constructive suggestions for improvement. Regarding your comment on the Mann-Whitney test and Tukey’s test in ANOVA, we have now clarified their specific applications in the manuscript and we made a mistake for writing “Mann-Whitney test” was used. Additionally, we have revised the figure legend to explicitly describe the use of these statistical tests in assessing the differences. The corresponding changes can be found at line 190-192, line 633-636, line 644, line 653 and line 671-672.

Thank you again for your valuable insights and constructive input, which has helped us improve the quality of our manuscript.

---

## [Decision Letter · Decision Letter 1]

26 Jun 2025

Impacts of cellulase and xylanase addition on antibiotic resistance and microbial community during dairy manure composting

PONE-D-24-53884R1

Dear Dr. Jin,

We’re pleased to inform you that your manuscript has been judged scientifically suitable for publication and will be formally accepted for publication once it meets all outstanding technical requirements.

Kind regards,

Shweta Sharma, PhD

Academic Editor

PLOS ONE

Additional Editor Comments (optional):

Reviewers' comments:

Reviewer's Responses to Questions

**Comments to the Author**

Reviewer #1: All comments have been addressed

2. Is the manuscript technically sound, and do the data support the conclusions?

Reviewer #1: Yes

3. Has the statistical analysis been performed appropriately and rigorously?

Reviewer #1: Yes

4. Have the authors made all data underlying the findings in their manuscript fully available?

Reviewer #1: Yes

5. Is the manuscript presented in an intelligible fashion and written in standard English?

Reviewer #1: Yes

Reviewer #1: (No Response)

**Do you want your identity to be public for this peer review?** For information about this choice, including consent withdrawal, please see our Privacy Policy

Reviewer #1: No

---

## [Editor Report · Acceptance letter]

PONE-D-24-53884R1

PLOS ONE

Dear Dr. Jin,

I'm pleased to inform you that your manuscript has been deemed suitable for publication in PLOS ONE. Congratulations! Your manuscript is now being handed over to our production team.

Kind regards,

on behalf of

Dr. Shweta Sharma

Academic Editor

PLOS ONE